# flDPnn: Accurate intrinsic disorder prediction with putative propensities of disorder functions

Gang Hu [1,4], Akila Katuwawala[2,4], Kui Wang[3,4], Zhonghua Wu[3], Sina Ghadermarzi[2], Jianzhao Gao [3] & Lukasz Kurgan [2✉]

Identification of intrinsic disorder in proteins relies in large part on computational predictors, which demands that their accuracy should be high. Since intrinsic disorder carries out a broad range of cellular functions, it is desirable to couple the disorder and disorder function predictions. We report a computational tool, flDPnn, that provides accurate, fast and comprehensive disorder and disorder function predictions from protein sequences. The recent Critical Assessment of protein Intrinsic Disorder prediction (CAID) experiment and results on other test datasets demonstrate that flDPnn offers accurate predictions of disorder, fully disordered proteins and four common disorder functions. These predictions are substantially better than the results of the existing disorder predictors and methods that predict functions of disorder. Ablation tests reveal that the high predictive performance stems from innovative ways used in flDPnn to derive sequence profiles and encode inputs. flDPnn's webserver is available at http://biomine.cs.vcu.edu/servers/flDPnn/

[1] School of Statistics and Data Science, LPMC and KLMDASR, Nankai University, Tianjin, China. [2] Department of Computer Science, Virginia Commonwealth University, Richmond, VA, USA. [3] School of Mathematical Sciences and LPMC, Nankai University, Tianjin, China. [4] These authors contributed equally: Gang Hu, Akila Katuwawala, Kui Wang. ✉email: lkurgan@vcu.edu

The intrinsically disorder regions (IDRs) in the protein sequences lack a stable structure under physiological conditions and carry out their functions often without fully folding[1–3]. Recent bioinformatics studies suggest that IDRs are abundant in nature, particularly in the eukaryotes where they are found in over 30% of proteins[4,5]. Moreover, proteins with IDRs are associated with pathogenesis of many human diseases[6], including cancers[7] and neurodegenerative disorders[8], motivating interest in utilizing these proteins as drug targets[9–11]. Since so far only several thousand IDRs were characterized experimentally[12], computational methods that predict these regions directly from protein sequences have emerged as a viable way to identify and functionally investigate IDRs[13–16]. These methods are applied to structurally and functionally characterize specific proteins and protein families[17,18] and to analyze prevalence and functions of disorder at the proteome scale[19–23]. Predictive quality of disorder predictors was evaluated in several comparative surveys[24–26] and community-driven assessments that include CASP (Critical Assessment of protein Structure Prediction), where they were run as a sub-category of the protein structure prediction, and more recently CAID (Critical Assessment of protein Intrinsic Disorder). The first community assessment of the intrinsic disorder predictions was in CASP5 and featured six predictors[27]. The disorder prediction evaluations were continued as part of the CASP experiments until CASP10 that covered 28 disorder predictors[28]. Following CASP10, the organization of the assessment was shifted to the disorder prediction community. The first CAID experiment was completed recently and featured a record-breaking collection of 32 predictors[29]. CAID has shown that several current tools, such as AUCpreD[30], ESpritz[31], RawMSA[32], SPOT-Disorder2[33], and SPOT-Disorder-Single[34], produce accurate disorder predictions with AUC (area under the ROC curve) of about 0.75 when evaluated on the CAID's test dataset[29] that relies on the experimental annotations from the popular DisProt database[12]. However, none of the current predictors has reached AUC > 0.78 in the CAID experiment and these methods predict intrinsic disorder without explaining what it does. This is a substantial drawback since IDRs carry out a broad spectrum of cellular functions. In particular, the inherent plasticity allows IDRs to interact with a wide range of different partners[35–38] and to serve as linkers that facilitate and regulate inter- and intra-domain movement and allosteric regulation[39,40]. While there are specialized methods that predict specific functions of IDRs, nearly all of them focus on the prediction of the protein-binding IDRs[16,41,42] and they are decoupled from the disorder predictions, i.e., their predictions could be misaligned with the outputs of the disorder predictors[43]. Besides the predictors of the protein-binding IDRs, there are only three methods that focus on other disorder functions including APOD[44] and DFLpred[45] that predict linkers and DisoRDPbind that predicts protein-binding, DNA-binding and RNA-binding IDRs[46,47].

Here, we developed a computational tool, flDPnn, that produced the most accurate predictions of disorder (AUC = 0.814) and the fully disordered proteins (i.e., proteins for which disorder covers at least 95% of their sequences) in CAID. Moreover, flDPnn generates putative functions for the predicted IDRs covering the four most commonly annotated functions, including protein-binding, DNA-binding, RNA-binding, and linkers[12,41]. Selection of these functions is motivated by the sufficient amount of the corresponding experimental data to train and test machine-learning models[12]. While CAID also benchmarked quality of the prediction of disordered binding regions, flDPnn did not participate in this assessment since its modules for the prediction of protein/DNA/RNA-binding and linkers were developed after the CAID experiment was concluded.

## Results

**Architecture of flDPnn.** The flDPnn's architecture implements a three-step prediction process (Fig. 1). First, the input sequence is

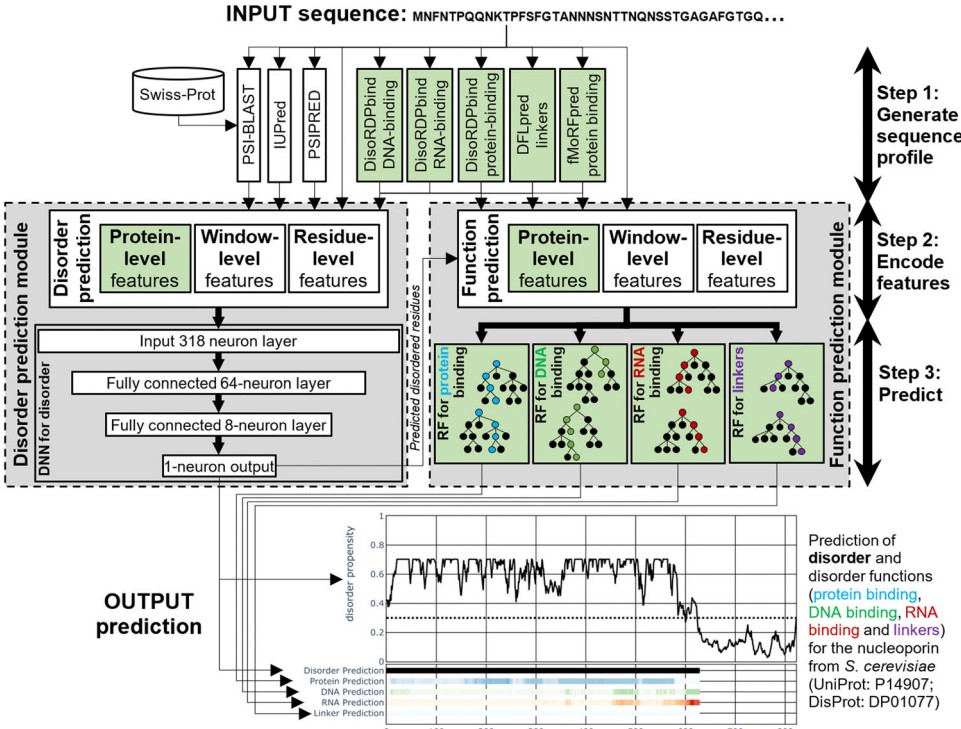

**Fig. 1 The architecture of the flDPnn disorder predictor.** Green highlights identify novel elements. Gray boxes denote the disorder prediction and the disorder function prediction modules. DNN (deep neural network); RF (random forest). The example input and outputs correspond to the prediction for the nucleoporin protein from *S. cerevisiae* (UniProt: P14907; DisProt: DP01077).

processed by several tools to produce relevant putative structural and functional information that forms the sequence profile. Second, the profile is used to encode numerical features. Third, the features are input into a machine-learning model that produces predictions. We introduced three key innovations into this architecture that touch each of the three steps (green highlights in Fig. 1). We extended the commonly used profiles to include putative disorder functions produced by DisoRDPbind[47], DFLpred[45], and fMoRFpred[48]. This is driven by the fact that intrinsic disorder carries a multitude of cellular functions that are associated with different sequence patterns[2], which calls for an equally diverse profile. Next, we encoded the features by aggregating the profile data at three levels: residue, window and protein levels. While current disorder predictors use the residue level and/or window-level encodings, we added the protein-level features that express the overall bias of a given sequence to be disordered or structured. Finally, we included a module for the function prediction that focuses specifically on the disordered regions generated by flDPnn. This way the function predictions line up with and complement the disorder predictions. We utilized a deep neural network for the disorder prediction and a random forest for the function prediction, motivated by the popularity of these machine-learning models in the disorder prediction field[30,32–34,49,50].

This architecture produces high-quality disorder predictions that are accompanied by the putative propensities for the four disorder functions. Figure 1 illustrates predictions produced by flDPnn for one of the test proteins that we utilized to assess flDPnn's performance, nucleoporin from *S. cerevisiae* (UniProt: P14907; DisProt: DP01077). These predictions include numeric disorder propensities, which quantify likelihood that a given residue in the input protein chain is disordered and binary prediction that categorizes each residue as either disordered or ordered. Nucleoporin is known to harbor a long IDR between positions 1 and 603 that interacts with protein partners[51,52]. flDPnn predicts disorder between positions 1 and 623 (black lines in the outputs shown in Fig. 1), however, the propensities at the c-terminus of this segment are lower suggesting smaller likelihood for disorder. The function predictions (color-coded lines in the outputs shown in Fig. 1, where intensity of color represent likelihood for a given function) suggest high likelihood for protein interactions (in blue), low likelihood for DNA interactions (in green) and linkers (in violet), and high likelihood for the RNA binding at the c-terminus of the putative IDRs. However, the last prediction should be considered to have lower quality given the relatively low likelihood for disorder in this region.

**Prediction of intrinsic disorder**. The flDPnn predictor was designed and trained before the CAID experiment on the data extracted from the DisProt 7.0 database[53]. We created a separate test dataset of 176 proteins from DisProt 7.0 that share low, <25% similarity to the training proteins that we used to develop our predictor. flDPnn was submitted to the CAID experiment for independent assessment. The CAID organizers collected and ran code for each submitted predictor on their own hardware using a community curated dataset of 646 proteins that were unknown to the authors of the predictors[29]. In CAID, the putative disorder propensities were evaluated with AUC while the binary predictions were assessed with F1 and MCC metrics. We detail the datasets and metrics in the Methods section.

We reproduced the results of the top 10 predictors from the CAID experiment in Fig. 2A. flDPnn secures AUC = 0.814, MCC = 0.370, and F1 = 0.483. The second-best predictor, flDPlr, is a derivative of the flDPnn model that uses the same architecture

where the deep neural network is replaced with a simple logistic regression model. The top-performers in CAID rely on more sophisticated models including RawMSA[32], ESpritz[31], DisoMine[54], SPOT-Disorder2[33], AUCpreD[30], and SPOT-Disorder-Single[34] that utilize deep convolutional and/or recurrent neural networks. The fact that flDPlr's results improve over the results of these models reveals that the predictive performance primarily depends on the quality of the input profile and features, not the sophistication of the predictive model. This finding is in line with a recent study similarly concludes that extraction of high-quality feature spaces leads to better results than those provided by deep models that automatically derive latent features spaces[55]. The CAID experiment also shows that the runtimes of the top 10 predictors differ by as much as three orders of magnitude; these measurements are in Fig. 2F in the CAID article[29]. The fast predictors, which produce results in under 1 min per protein, include flDPnn (5 to 10 s per protein), flDPlr (5 to 10 s), ESpritz (1 s), SPOT-Disorder-Single (20 to 40 s), DisoMine (1 s) and AUCpredD-np (1 s). The slower methods, such as SPOT-Disorder2, RawMSA, AUCpredD and Predisorder[56], take several to dozens of minutes to complete the prediction.

Moreover, we compared flDPnn with the similarly fast disorder predictors on the test dataset (Fig. 2B). This group of the fast methods includes SPOT-Disorder-Single and ESpritz-D that performed well in CAID and two versions of the popular IUPred2A method: IUPred2A-long and IUPred2A-short[57]. flDPnn generates higher quality predictions with AUC = 0.839, MCC = 0.491 and F1 = 0.626 when compared to the second-best ESpritz-D with AUC = 0.799, MCC = 0.428, and F1 = 0.603. The corresponding ROC curves reveal that the improvements are consistent over the entire range of the false-positive rates (Fig. 2C). Similar to the disorder assessment at CASP[28], we evaluated statistical significance of these differences by resampling half the test dataset 10 times and comparing results for each pair of the five disorder predictors. This evaluates whether the improvements offered by the better of the two compared methods are robust to different datasets. We used paired *t*-test if the measured values were normal (we assessed normality with the Anderson-Darling test at 0.05 significance); otherwise we applied the Wilcoxon test. The resulting p-values are listed in Supplementary Table 1 and show that the AUC and F1 scores produced by flDPnn are statistically significantly better than the results of the other four methods (*p*-value < 0.05).

**Prediction of fully disordered proteins**. Figure 2D, E evaluate predictions of the fully disordered proteins in the CAID experiment and on the test dataset, respectively. Here, the task is to differentiate between the non-fully disordered proteins and fully disordered proteins, where the latter are defined as proteins with over 95% of disordered residues. These proteins are of particular interest since they are virtually impossible to solve structurally via X-ray crystallography and were suggested to encode for a set of specialized protein functions[58,59]. flDPnn provides the most accurate prediction of the fully disordered proteins, measured with both MCC and F1, on the CAID dataset[29] (Fig. 2D).

The assessment on the test dataset similarly reveals that flDPnn outperforms the four fast predictors (Fig. 2E). flDPnn predicts the fully disordered proteins with MCC = 0.63 and F1 = 0.67 when compared to the second-best SPOT-Disorder-Single that secures MCC = 0.58 and F1 = 0.57. The results of the resampling tests that evaluate significance of the differences are given in Supplementary Table 1. They demonstrate that the improvements on the test dataset offered by flDPnn are statistically significant when compared to the other four predictors (*p*-value > 0.05), with

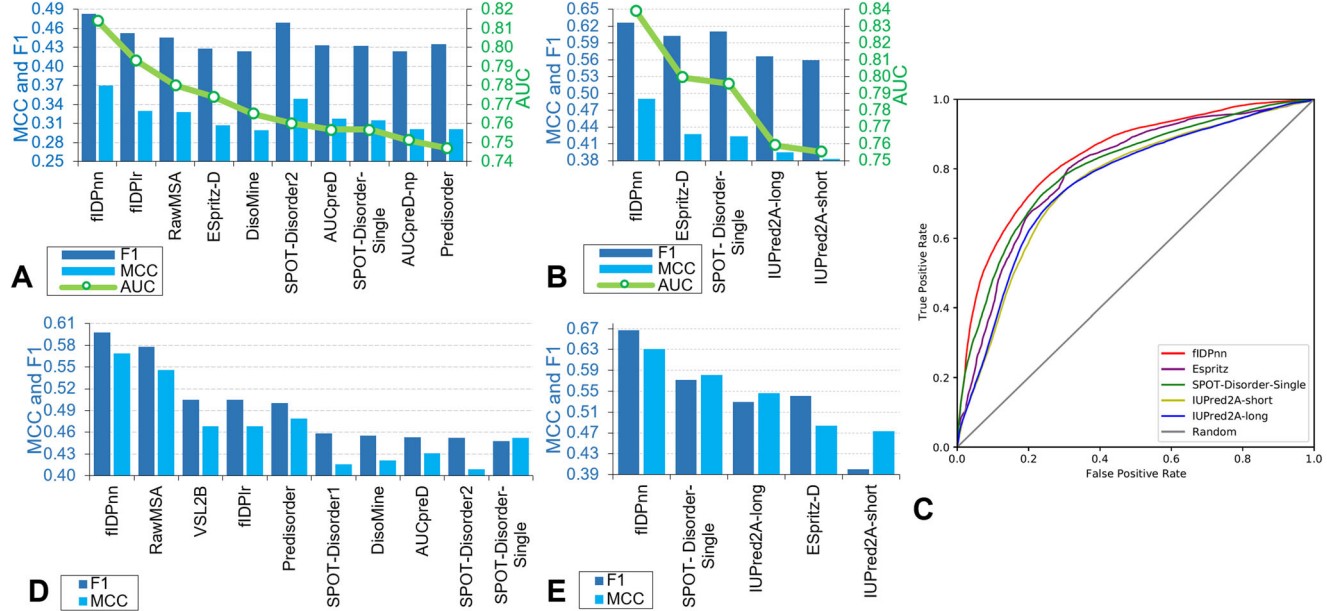

**Fig. 2 Comparison of predictive performance between flDPnn and other disorder predictors.** Assessment of the disorder predictions in the CAID experiment (**A**) and on the test dataset (**B** and **C**) and the predictions of the fully disordered proteins in the CAID experiment (**D**) and on the test dataset (**E**). AUC (area under the ROC curve); MCC (Matthews correlation coefficient). The predictive quality of the putative disorder propensities is quantified with AUC (green lines) and ROC curves (**C**). The ROC curves are color-coded where flDPnn is shown in red, ESpritz in violet, SPOT-Disorder-Single in dark green, IUPred2A-short in light green, IUPred2A-long in blue and random predictor in gray. The quality of the binary disorder predictions is assessed with MCC and F1 scores (blue bars). Panels **A** and **D** are reproduced from Fig. 2, Table 1, and Supplementary Table 5 from the CAID publication[29].

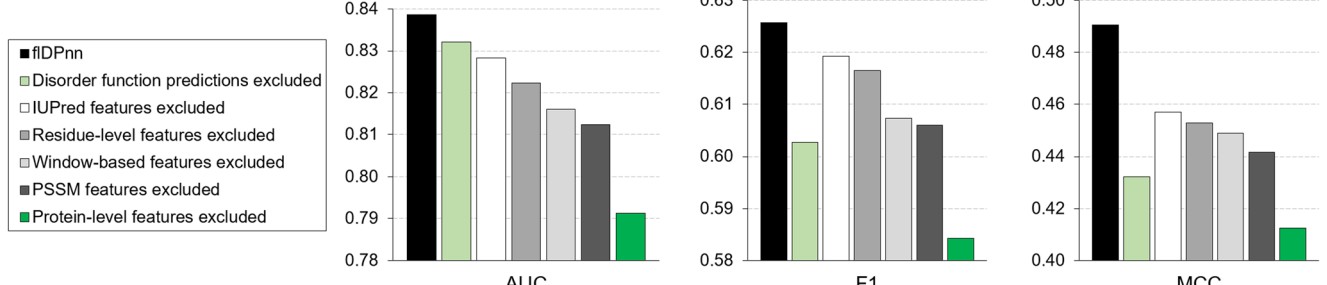

**Fig. 3 Ablation analysis of the flDPnn predictor on the test dataset.** Black bars show results for the flDPnn model. Green bars quantify predictive quality where one of the two major innovations (disorder functions predictions in the profile and protein-level feature encoding) is excluded. Gray bars show the performance where one of the architectural elements that are often utilized by the current predictors (residue-level and window-level feature encoding) is excluded. AUC (area under the ROC curve); MCC (Matthews correlation coefficient).

the exception of MCC when compared to SPOT-Disorder-Single ($p$-value = 0.16).

**Ablation analysis**. The favorable predictive quality of flDPnn can be traced to its architecture. The main determinants of the predictive performance are the scope and formulation of the sequence profile and the feature encodings (Fig. 1). We performed ablation analysis where we observed the impact of the exclusion of the key elements of the profile and feature encodings on the results on the test dataset (Fig. 3). This covers exclusion of the two novel components, disorder functions predictions in the sequence profile and protein-level feature encodings (green bars in Fig. 3). We also considered elimination of the window-level and the residue-level encodings (gray bars in Fig. 3) and the PSI-BLAST and IUPred generated inputs that are often utilized by current predictive tools. We omit each of these elements separately to measure their individual contribution to the flDPnn model. We detail the corresponding six ablation configurations in Supplementary Table 2. Results in Fig. 3 reveal that the biggest

drop in performance is associated with the removal of the protein-level feature encodings, from 0.839 to 0.791 in AUC and from 0.490 to 0.412 in MCC. Moreover, exclusion of each of the six elements results in a lower predictive quality. Based on resampling the test dataset, we found that the reductions in the AUC when comparing flDPnn with the six ablation experiments are statistically significant ($p$-value < 0.05). Moreover, the decreases in F1 and MCC scores are also statistically significant ($p$-value < 0.05), except when comparing flDPnn to the setup where IUPred generated features are excluded ($p$-value = 0.14 for F1 and 0.09 for MCC). The latter suggests that deletion of IUPred may sometimes not result in worsening the predictive performance. The complete set of $p$-values is available in the Supplementary Table 3. Overall, these results suggest that the two innovations that we introduced substantially contribute to the high accuracy of the flDPnn's predictions.

**Prediction of disorder functions**. flDPnn is the only predictor that provides putative functions for the predicted IDRs. While the

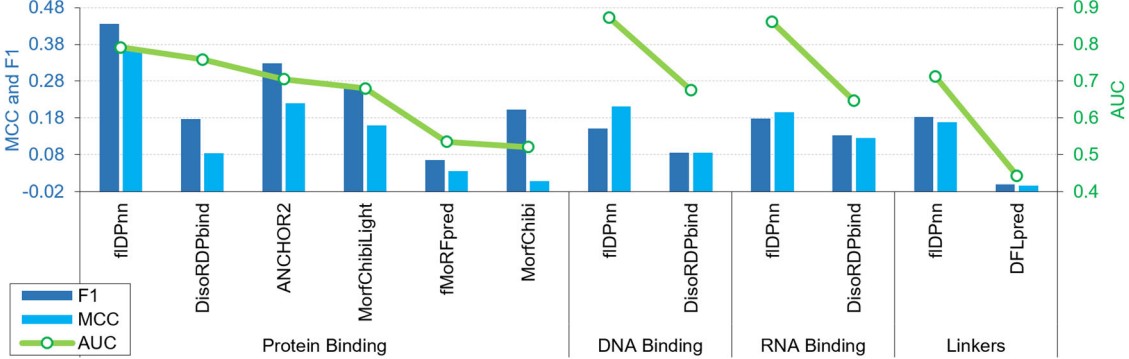

**Fig. 4 Assessment of the quality of the disorder function predictions for the IDRs predicted by flDPnn on the test dataset.** AUC (area under the ROC curve); MCC (Matthews correlation coefficient). The predictive quality for the putative function propensities and binary predictions is quantified with AUC (green lines) and with MCC and F1 scores (blue bars), respectively.

recently released DEPICTER resource facilitates combining predictions of the disorder and disorder functions[43], these results are produced by several different methods resulting in potentially conflicting predictions. For instance, putative protein-binding IDRs produced by one of the disorder function predictors could be predicted as structured by the disorder predictor. In contrast, flDPnn utilizes random forest models to predict the putative functions for the IDRs generated by the deep network model, ensuring that the predicted functions align with the putative IDRs (Fig. 1). We assessed the predictive performance for the four disorder functions predicted by flDPnn on the test dataset (Fig. 4). We compared the flDPnn's predictions against the state-of-the-art disorder function predictors: DisoRDPbind[47], ANCHOR2[57], MoRFChibiLight[60], fMoRFpred[48], and MoRFChibi[60] for the prediction of the protein-binding IDRs; DisoRDPbind, the sole predictor of the DNA and RNA-binding IDRs; and DFLpred[45] for the prediction of linkers. This group of methods includes the top three methods based on the CAID results concerning predictions of the disordered binding regions: ANCHOR2, DisoRDPbind and MoRFChibiLight. The quality of the flDPnn's function predictions on the test dataset quantified by AUC, F1 and MCC metrics is better than the quality of the predictions by the current tools. flDPnn secures AUC of 0.79, 0.87, 0.86 and 0.71 for the prediction of the protein-binding, DNA-binding, RNA-binding and linkers, respectively. The ROC curves for flDPnn and the second-best predictor, selected according to the AUC scores in Fig. 4, are compared in Supplementary Fig. 1. Similar to the assessment of the disorder predictions, ROC curves show that flDPnn maintains higher levels of predictive performance over the entire spectrum of the false-positive rates. Moreover, based on the resampling analysis, flDPnn offers statistically significant improvements in AUC when compared to the results of DFLpred, DisoRDPbind and the five predictors of the disordered protein-binding IDRs ($p$-value < 0.05). The flDPnn's F1 scores are also significantly better for the prediction of the protein-binding and linker regions ($p$-value < 0.05), while the differences in F1 for the prediction of the nucleic-acid binding regions are not statistically significant; $p$-value = 0.12 for DNA binding and 0.08 for RNA-binding when flDPnn is compared to DisoRDPbind. The complete list of the $p$-values is in Supplementary Table 4.

## Discussion

We developed flDPnn, state-of-the-art predictor of disorder and disorder functions from protein sequences. Extensive empirical assessment in the CAID experiment and on an independent and low-similarity test dataset show that flDPnn offers accurate predictions of disorder, fully disordered proteins and four common

disorder functions. Ablation analysis suggests that the high predictive quality stems from the innovations applied to the predictive inputs, including extended sequences profile and protein-level feature encoding. This finding is supported by the fact that the difference between the results of flDPnn and flDPlr models in CAID is not very large[29]. These models use the same inputs and differ only in the predictive model that they utilize, i.e., deep neural network in flDPnn vs. logistic regression in flDPlr. This suggests that the type of the model used has relatively modest influence on the predictive performance. Another key advantage of flDPnn is the use of runtime-efficient tools to derive the inputs. In particular, we used PSI-BLAST search on the small SwissProt database and fast IUPred and the single-sequence version of PSIPRED to achieve a good compromise between speed and accuracy. Consequently, the entire prediction process that produces disorder and disorder function predictions takes only about 5 to 10 s per protein, making flDPnn one of the fastest disorder predictors[29]. This fact was highlighted in the recent article that summarizes CAID results where the authors noted flDPnn is at least an order of magnitude faster than its close competitors[61]. In the nutshell, flDPnn has pushed beyond the limits of the current disorder and disorder function predictors to provide accurate, fast and comprehensive collection of disorder and disorder function predictions.

We provide standalone code for flDPnn at https://gitlab.com/sina.ghadermarzi/fldpnn and a docker container at https://gitlab.com/sina.ghadermarzi/fldpnn_docker. Moreover, we also offer a convenient webserver version of flDPnn at http://biomine.cs.vcu.edu/servers/flDPnn/. We plan to further extend the scope to other disorder functions, such as lipid, metal and small molecule interactions, when the amount of the experimental data becomes sufficient to train and test these models.

## Methods

**Datasets**. The flDPnn predictor was designed, trained and tested before CAID on the 745 experimentally annotated proteins from the DisProt 7.0 database[53]. We divided these proteins at random into three disjoint datasets: the training dataset (445 proteins) to train the machine-learning models, the validation dataset (100 proteins) to set hyper-parameters of the trained models, and the test dataset (200 proteins) to compare performance with the current disorder and disorder function predictors. Next, we clustered the combined set of 200 test and 445 training proteins using the CD-HIT algorithm with 25% sequence similarity[62]. We removed the test proteins that are in clusters that include any of the training proteins. The resulting test dataset includes the remaining 176 test proteins that consequently share <25% sequence similarity to the training proteins. This was motivated by a recent study that shows that lack of the sequence similarity reduction results in overestimation of the predictive performance due to the potential overfitting[24]. flDPnn participated in the CAID experiment where 32 disorder predictors were evaluated on a blind set of 646 proteins. Experimental data for these proteins is consistent with the DisProt format and was unavailable to the developers of these predictors. However, these proteins were not filtered for their similarity to the

training datasets of the participating disorder predictors. Similar to CAID and other comparative assessments[25,26], amino acids in these test proteins were labeled as structured unless they were experimentally annotated as disordered. Disorder function annotations were extracted from DisProt[12,53]. The CAID dataset is available on the DisProt website at https://www.disprot.org/. The training, validation and test datasets are available at http://biomine.cs.vcu.edu/servers/flDPnn/.

**Assessment of predictive performance**. Disorder and disorder function predictors produce two outputs: real-valued propensity that quantifies likelihood that a given amino acid is disordered or has a given disorder function and a corresponding binary classification (disordered vs. structured/does vs. does not have a given disorder function). Typically, the binary predictions are generated from the propensities such that residues with propensities greater than a given threshold are predicted as disordered/functional, and otherwise they are predicted as structured/non-functional. We evaluate the predictive performance of the binary predictions with the two metrics and thresholds that are consistent with the CAID experiment[29]: F1 and MCC (Matthews correlation coefficient). MCC ranges between −1 and 1, where −1 denotes an inverted prediction (all predictions are flipped compared to the experimental values), 0 denotes a random result and 1 denotes a perfect prediction. F1 ranges between 0 and 1 where higher value denotes more accurate prediction. Consistent with other assessments including CAID[14,25,26,28,29], we used the area under the receiver-operating characteristic curve (ROC-AUC) to evaluate predictive quality of the propensities. The ROC curve is a relation between true-positive rates (TPRs) and false-positive rates (FPRs) that is computed by thresholding the propensities where the thresholds are the set of all unique propensities produced by a given predictor. ROC-AUC ranges between 0.5 (equivalent to a random prediction) and 1 (perfect prediction). F1, MCC, TPR and FPR are defined in the Supplementary Table 5.

**Design and training of flDPnn**. flDPnn generates predictions in three steps (Fig. 1): sequence profile, feature encoding and prediction using a machine-learning model. The profile covers putative structural and functional information relevant to the disorder and disorder function predictions. We extracted this information from the input protein chains utilizing popular and fast bioinformatics tools. The profile includes commonly used in the disorder prediction field secondary structure predicted with the single-sequence version of PSIPRED[63], disorder predicted with IUPred[64], positions specific scoring matrix (PSSM) generated with PSI-BLAST[65] from the SwissProt database, and the entropy-based conservation scores computed from PSSM[66]. We utilized the putative disorder produced by IUPred, one of the most popular disorder predictors, as a baseline that we refine with the help of the other inputs and the deep network model. We also introduced innovative elements into the profile including the putative disordered DNA and RNA binding generated with DisoRDPbind[47], putative disordered protein binding produced by DisoRDPbind and fMoRFpred[48], and disordered linkers predicted by DFLpred[45].

We encoded the profile into three feature sets that focus on the residue-level, window-level and protein-level information. The residue-level and window-level information is extracted with the popular sliding window approach, where the residue in the middle of a small sequence segments is predicted based on the information from all residues in that segment. The residue-level features cover the profile values for individual residues in a small window of five residues for the disorder prediction and one residue for the disorder function prediction. The window-level features aggregate the profile information by computing average over a larger window of 15 residues for the disorder prediction and 11 residues for the disorder function prediction. We also introduce new protein-level encoding, which quantifies the overall bias of a given sequence to be disordered or to have functional regions. The protein-level features include the sequence-average of the profile values, sequence length, and the distance to each sequence terminus.

We used deep feedforward neural network to predict disorder. Selection of this machine-learning algorithm was motivated by its popularity in the disorder prediction field[30,32–34]. The input layer of 318 nodes (which equals to the number of features) is followed by a dropout layer with 0.2 dropout rate, hidden layer with 64 nodes, dropout layer with 0.2 dropout rate, second hidden layer with 8 nodes, and the output layer with one node that produces the disorder propensities. The gradually smaller layers simulate progressive reduction of the latent feature spaces that eventually condense to the propensity score. We included the dropout layers to prevent overfitting into the training dataset[67]. We used ReLU activation functions for all nodes except for the output node where we applied the sigmoid function to properly scale the propensities. We empirically selected the hyper-parameters including number of layers and the number of nodes in the hidden layer by a grid search where the networks trained on the training were optimized to maximize AUC on the validation dataset. We also experimented with a popular shallow machine-learning algorithm, logistic regression. The corresponding flDPlr model, which uses the same profile and features as the deep neural network-based flDPnn, was evaluated in CAID and provided lower quality predictions: AUC = 0.814 for the deep network vs. 0.793 for logistic regression[29].

We used the random forest (RF) algorithm to predict disorder functions. The amount of the training data (which cover the putative IDRs generated by the deep network) was too small to train deep networks. The choice of RF was motivated by its success in related disorder function predictors[49,50]. We used grid search to empirically optimize RF hyper-parameters including number of trees and tree depth to maximize AUC computed based on the 3-fold cross-validation on the training dataset. We opted to perform cross-validation rather than using the validation dataset since this test protocol is more robust to overfitting and training of RF is sufficiently fast. We repeated this parametrization for each of the four disorder functions. These results are in Supplementary Fig. 2. The training, validation and deployment of the predictive models were implemented with Python 3 (3.8.5) including the following packages: scikit-learn (0.23.2), keras (2.4.3), tensorflow (2.4.1) and pandas (1.2.2).

**Webserver**. flDPnn is available as a webserver at http://biomine.cs.vcu.edu/servers/flDPnn/. The webserver supports batch predictions of up to 20 FASTA-formatted protein sequences per request. The requests are placed into a queue that services multiple webservers and ensures load balancing between users. The prediction process is automated and performed on the server side. The server outputs numeric propensities for disorder and disorder functions together with corresponding binary predictions for each residue in the input protein chain(s). These results are delivered in the browser window via a unique URL. The users are informed by email (if an email address was provided) when the prediction completes. The results are available in three formats: (1) a parsable csv-formatted file; (2) an interactive graphical format that we developed with plotly (4.14.3); and (3) a png image file generated from the graphic. The graphical format allows the users to select/deselect specific predictions, identify propensity scores, residue type and position on mouse hover, and zoom on a specific sequence region.

**Reporting summary**. Further information on research design is available in the Nature Research Reporting Summary linked to this article.

## Data availability
Source data are provided with this paper. Its was collected by parsing the publicly available DisProt repository (https://www.disprot.org/). This includes the training, validation, test and CAID datasets. Other data are available from the corresponding author upon reasonable request. In particular, the parsed datasets (including raw data and identifiers) are freely available (no restrictions) at http://biomine.cs.vcu.edu/servers/flDPnn/. We also utilize the publicly available SwissProt dataset (https://www.uniprot.org/statistics/Swiss-Prot) to produce PSSM. Source data are provided with this paper.

## Code availability
We release code for flDPnn at https://gitlab.com/sina.ghadermarzi/fldpnn. We also provide a docker container that simplifies local installation at https://gitlab.com/sina.ghadermarzi/fldpnn_docker.

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

## Acknowledgements

This work was supported in part by the National Science Foundation (1617369) and the Robert J. Mattauch Endowment funds to L.K., National Natural Science Foundation of China (31970649) to G.H., K.W. and Z.W., and National Natural Science Foundation of China (11701296) to J.G.

## Author contributions

L.K. conceptualized and administered the study. G.H., K.W., Z.W., J.G. and L.K. designed and implemented the deep network predictor. A.K. and L.K. designed and implemented the random forest predictors. G.H., K.W., Z.W. and A.K. collected and annotated datasets. All authors carried out experiments and analyzed results. A.K. and S.G. implemented and deployed the webserver. L.K. drafted the manuscript. All authors revised and approved the final draft of the manuscript.

## Competing interests

The authors declare no competing interests.
