## [Peer Review File · Nature Communications]

REVIEWER COMMENTS

Reviewer #1 (Remarks to the Author):

The manuscript by Hu et al. describes fIDPnn, a novel predictor for intrinsic disorder (ID) in protein sequences along with four main functional categories. Indeed, the method performed well in the CAID blind test recently (i.e. last week) published in Nature Methods, although the differences between methods in the top-performing group were not really statistically significant. The main innovation, apart from a state-of-the-art performance, rests in the inclusion of ID function prediction in the same tool. Intriguingly, the contribution of the neural network (fIDPnn) over a simple linear regression (fIDPIr) in CAID are rather limited. This would be worth exploring in more detail, as it suggests the data used for training and the input features are key to the performance. This being said, the manuscript reads overall well. It raises the following concerns/questions:

1. Contribution of input features. The authors use both a PSI-BLAST profile and IUpred, a fast ID predictor based on biophysical principles, as input. These should be addressed separately in ablation studies, to establish their precise contribution.

a) It is a bit strange to have an ID predictor as input for another ID predictor. The authors should justify this choice or otherwise spell out clearly that they are proposing a meta-predictor.

b) Regarding the PSI-BLAST profile, is this used only to extract conservation or are the authors also searching directly for previous DisProt entries?

c) On a similar note, it is not clear what exactly are the components addressed in the ablation study. Here, the authors should provide a table with the correspondence between input features and groups in the ablation study.

2. Datasets. The authors make extensive use of the CAID results, which is generally correct, except perhaps for statistical significance (see below). The test set being used however raises a few questions. If this reviewer understands correctly, this is a random sampling of the DisProt 7.0 database. Did the authors make any provisions regarding maximum sequence identity between the splits? This point appears crucial to avoid overfitting on the data, which would put other methods at a disadvantage. Indeed, is this a single partition of the DisProt dataset? What happens if 10-fold crossvalidation is used instead?

3. Statistical significance. The authors claim that their method is statistically significantly better than others. Yet on the CAID dataset, the official assessment says that the top performing methods (including fIDPnn) cannot be reliably distinguished. A full p-value matrix like the one in the CAID supplementary material should be provided to support this claim. Otherwise it should be toned down.

4. Discussion. This appears very short. The authors should make an effort to put their method into the broader context of ID predictions. E.g. the difference between fIDPnn and fIDPIr is not very large, suggesting that the input features are important and well selected. Moreover, the authors have made a few interesting choices (e.g. PSI-BLAST search on SwissProt) that make for a good compromise between speed and accuracy. All of this would be worth discussing.

Reviewer #2 (Remarks to the Author):

This article describes fIDPnn, a new system for the prediction of intrinsic protein disorder and some prediction of disorder function. The main predictor is a relatively simple deep architecture, but relies on careful design of input features, which in this case seems to be particularly helpful because high-quality curated training datasets are relatively small (hundreds of proteins, rather than the several thousands of even tens of thousands that would be available for structural predictions). In the same vein, disorder function prediction, which relies on even smaller training sets, is performed via random forests. All these are very practical design decisions which seem completely logical given the data availability circumstances. Considerably strengthening the article, fIDPnn was entered into CAID, a blind disorder prediction competition which is run and evaluated independently of the authors, and by most measures it did outperform the other competitors, so there is a reasonable claim made in the article that the system presented is the absolute state of the art in the field. An ablation study is also presented which seems to suggest that the features utilised do indeed contribute to the success of the system, and that using a deep network as the main component of the disorder predictor also is advantageous compared to less sophisticated methods. The method is also freely available as a web server.

As a relatively minor note, while I commend the release of their data and availability of the web server, I would strongly encourage the authors to release their method as standalone software to maximise its impact.

Reviewer #1

Comment 1: *The manuscript by Hu et al. describes fIDPnn, a novel predictor for intrinsic disorder (ID) in protein sequences along with four main functional categories. Indeed, the method performed well in the CAID blind test recently (i.e. last week) published in Nature Methods, although the differences between methods in the top-performing group were not really statistically significant. The main innovation, apart from a state-of-the-art performance, rests in the inclusion of ID function prediction in the same tool. Intriguingly, the contribution of the neural network (fIDPnn) over a simple linear regression (fIDPlr) in CAID are rather limited. This would be worth exploring in more detail, as it suggests the data used for training and the input features are key to the performance. This being said, the manuscript reads overall well.*

REPLY: Thank you.

Comment 2: *Contribution of input features. The authors use both a PSI-BLAST profile and IUPred, a fast ID predictor based on biophysical principles, as input. These should be addressed separately in ablation studies, to establish their precise contribution.*

REPLY: We include the two additional ablation experiments where we remove features that are generated based on the outputs generated by PSI-BLAST and IUPred. We correspondingly extended the scope of Figure 3 and revised the discussion of this figure in text (see the “Ablation analysis” section). We note that the results are based on the new similarity-reduced test dataset (established in response to your comment 6) and thus all the numbers are slightly different than in the previous version of this figure. The original conclusions were not affected by these additions/changes – we still show that each of the six tested elements provides an important contribution and that, most importantly, our innovations help to raise accuracy of fIDPnn.

Comment 3: *It is a bit strange to have an ID predictor as input for another ID predictor. The authors should justify this choice or otherwise spell out clearly that they are proposing a meta-predictor.*

REPLY: The use of ID predictor as an input to predict ID is done frequently in this field. A common view is that meta-models combine results of multiple predictors of ID (sometimes without using any other inputs) while methods that use a single predictor as input perform refinement, not a meta-prediction. We now briefly explain that in the “Design and training of fIDPnn” section as follows: “*We utilized the putative disorder produced by IUPred, one of the most popular disorder predictors, as a baseline that we refine with the help of the other inputs and the deep network model.*”

Comment 4: *Regarding the PSI-BLAST profile, is this used only to extract conservation or are the authors also searching directly for previous DisProt entries?*

REPLY: As the text explains in the “Design and training of fIDPnn” section, we use PSI-BLAST to produce PSSM (using quick search on the Swiss-Prot dataset) which we use to generate the conservation scores. Figure 1 also identifies the Swiss-Prot dataset as the input for PSI-BLAST. We do not use PSI-BLAST to search the DisProt sequences.

Comment 5: *On a similar note, it is not clear what exactly are the components addressed in the ablation study. Here, the authors should provide a table with the correspondence between input features and groups in the ablation study.*

REPLY: Another good suggestion. We now provide a new Supplementary Table S2 (in the “Source Data.xlsx” file) that defines the six setups for the ablation analysis. We refer to this table in the “Ablation analysis” section.

Comment 6: Datasets. The authors make extensive use of the CAID results, which is generally correct, except perhaps for statistical significance (see below). The test set being used however raises a few questions. If this reviewer understands correctly, this is a random sampling of the DisProt 7.0 database. Did the authors make any provisions regarding maximum sequence identity between the splits? This point appears crucial to avoid overfitting on the data, which would put other methods at a disadvantage. Indeed, is this a single partition of the DisProt dataset? What happens if 10-fold cross validation is used instead?

REPLY: We originally did not screen the test proteins for their similarity to the training proteins. This was motivated by the fact that CAID does not do that either. However, we see value in this exercise. Correspondingly, we filtered the test proteins to remove those that share over 25% sequence similarity to the training proteins. We replaced the original test set of 200 proteins with the new, similarity-reduced test dataset that includes 176 test proteins (24 of the original test proteins were similar). We opted not to perform the 10-fold cross validation given the huge associated computational cost related to training the deep neural net. We note that we utilize dropout that should take care of overfitting and that test data is never used to optimize the network. We re-run all results using the similarity-reduced test dataset including Figures 2B, 2C, 2E, 3 and 4. We note that while the results went down by about 1%, this seemed to affect all tested methods. Consequently, the observations and conclusions were not affected by the change to the test dataset. We explain the procedure to collect the similarity-reduced test dataset in the “Datasets” section and we also mention the use of this new test dataset at the beginning of the “Prediction of intrinsic disorder” section. We also update discussion of the results from the abovementioned Figures to the new results on the similarity-reduced test dataset in the “Results” section.

Comment 7: Statistical significance. The authors claim that their method is statistically significantly better than others. Yet on the CAID dataset, the official assessment says that the top performing methods (including fIDPnn) cannot be reliably distinguished. A full p-value matrix like the one in the CAID supplementary material should be provided to support this claim. Otherwise it should be toned down.

REPLY: CAID covers a broad group of predictors that includes slow tools. While fIDPnn provides the best predictive performance in CAID, you are correct that the difference compared to some of the slow tools is not statistically significant. The statistical significance that we assess concerns comparative results on the test dataset and covers the collection of several fast predictors – those similar to the runtime to fIDPnn. We explain that in the “Prediction of intrinsic disorder” section. Moreover, we provide better separation between the discussion of the CAID results and the test set results to dispel potential confusion about the scope of the significance analysis – they are now located in separate paragraphs in sections “Prediction of intrinsic disorder” and “Prediction of fully disordered proteins”. We also extend the coverage of the fast methods from the original list of three that we included in the initial submission to four in this revision by adding IUPred2A-long. Per your request, we provide a full list of the p -values for each pair of the predictors that we compare on the test dataset. New Suppl. Table 1 gives p -values for the prediction of disordered residues and fully disordered proteins (this corresponds to

results in Figures 2B and 2E). New Suppl. Table S3 provides p -values for the ablation results from Figure 3. New Suppl. Table S4 lists p -values for the assessment of the disorder function predictions from Figure 4. We note that these results support our claim that fIDPnn provides the most accurate results, which in most cases are statistically significantly better than the results of the other tools that we utilize in the comparative study on the test dataset. We correspondingly revised discussions of the results in the “Results” section.

Comment 8: *Discussion. This appears very short. The authors should make an effort to put their method into the broader context of ID predictions. E.g. the difference between fIDPnn and fIDPlr is not very large, suggesting that the input features are important and well selected. Moreover, the authors have made a few interesting choices (e.g. PSI-BLAST search on SwissProt) that make for a good compromise between speed and accuracy. All of this would be worth discussing.*

REPLY: Thank you for this suggestion. We agree that the original version of the discussion section was short and rather cryptic. We extended this section with several observations related to the importance of the inputs features and speed of predictions. We support the latter with a citation to a recent opinion article on the CAID results (<https://www.nature.com/articles/s41592-021-01123-5>) that highlights advantages of fIDPnn, in particular its speed and predictive quality.

Reviewer #2

Comment 1: *This article describes fIDPnn, a new system for the prediction of intrinsic protein disorder and some prediction of disorder function. The main predictor is a relatively simple deep architecture, but relies on careful design of input features, which in this case seems to be particularly helpful because high-quality curated training datasets are relatively small (hundreds of proteins, rather than the several thousands of even tens of thousands that would be available for structural predictions). In the same vein, disorder function prediction, which relies on even smaller training sets, is performed via random forests. All these are very practical design decisions which seem completely logical given the data availability circumstances. Considerably strengthening the article, fIDPnn was entered into CAID, a blind disorder prediction competition which is run and evaluated independently of the authors, and by most measures it did outperform the other competitors, so there is a reasonable claim made in the article that the system presented is the absolute state of the art in the field. An ablation study is also presented which seems to suggest that the features utilised do indeed contribute to the success of the system, and that using a deep network as the main component of the disorder predictor also is advantageous compared to less sophisticated methods. The method is also freely available as a web server.*

REPLY: Thank you.

Comment 2: *As a relatively minor note, while I commend the release of their data and availability of the web server, I would strongly encourage the authors to release their method as standalone software to maximise its impact.*

REPLY: Many thanks for your advice. We were approached by several people asking for the standalone version of fIDPnn after the CAID article was published. We enthusiastically complied and released fIDPnn in two ways to further popularize our solution: as code and as a docker

container. The latter was a result of a few users having difficulty installing fIDPnn due to the number of other tools that need to be installed. We now explain the availability of both options in the “Discussion” and “Webserver and Code” sections in the revised article (with direct links) and provide the same links on the webserver site.

REVIEWER COMMENTS

Reviewer #1 (Remarks to the Author):

The manuscript by Hu et al. has been significantly improved and this reviewer feels that his previous points have been well addressed. A single point remains to avoid confusing the casual reader. CAID has benchmarked disordered binding regions, as described in the publication. This should be spelled out clearly in the current manuscript, along with the note that fIDPnn did not take part in the CAID evaluation as it was developed later. (Of course, CAID only benchmarked protein binding regions and not the other types in this round)

-- Silvio Tosatto

Reviewer #1

Comment 1: *The manuscript by Hu et al. has been significantly improved and this reviewer feels that his previous points have been well addressed.*

REPLY: Thank you.

Comment 2: *A single point remains to avoid confusing the casual reader. CAID has benchmarked disordered binding regions, as described in the publication. This should be spelled out clearly in the current manuscript, along with the note that fIDPnn did not take part in the CAID evaluation as it was developed later. (Of course, CAID only benchmarked protein binding regions and not the other types in this round)*

REPLY: Good point. We explain this at the end of the Introduction section. We say: “While CAID also benchmarked quality of the prediction of disordered binding regions, fIDPnn did not participate in this assessment since its modules for the prediction of protein/DNA/RNA-binding and linkers were developed after the CAID experiment was concluded.”